# First description and comparison of the morphological and ultramicro characteristics of the antennal sensilla of two fir longhorn beetles

**Zishu Dong**[1], **Fugen Dou**[2], **Yubin Yang**[2], **Jacob D. Wickham**[3], **Rong Tang**[1], **Yujing Zhang**[1], **Zongyou Huang**[1], **Xialin Zheng**[1], **Xiaoyun Wang**[1], **Wen Lu**[1]*

**1** Guangxi Key Laboratory of Agric-Environment and Agric-Products Safety, College of Agriculture, Guangxi University, Nanning, Guangxi, Peoples R China, **2** Texas A&M AgriLife Research Center, Beaumont, Texas, United States of America, **3** Chinese Academy of Sciences, Institute of Zoology, State Key Lab Integrated Management Pest Insects, Beijing, Peoples R China

* luwenlwen@163.com

**Data Availability Statement:** All relevant data are within the manuscript and its Supporting Information files.

## Abstract

*Allotraeus asiaticus* Schwarzer and *Callidiellum villosulum* Fairmaire are repeatedly intercepted in wood and wood products all over the world. As two common stem borers of *Cunninghamia lanceolata* (Lambert) Hooker, to further understanding of the differences in their living habits, behaviors and the mechanism of insect-host chemical communication, we observed the external morphology, number and distribution of antennal sensilla of *A. asiaticus* and *C. villosulum* with scanning electron microscopy (SEM), respectively. The results showed that 1st-5th subsegments of the flagellum are spined endoapically in *A. asiaticus* which is different from the previous report (1st-3rd of the flagellomere). Meanwhile, there were five subsegments on the flagellum of *C. villosulum* that were clearly specialized as serrated shapes on the 4th-8th flagellomeres. Four types (ten subtypes) of sensilla were both found on the antennae of these two fir longhorn beetles, named Böhm bristle (Bb), sensilla trichodea (ST I and II), sensilla basiconica (SB I, II and III), sensilla chaetica (SCh I, II, III and IV). There is one additional kind of morphological type of sensilla found on the antennae of *C. villosulum* compared to *A. asiaticus* which was related to their habit of laying eggs only on dry and injured fir branches, named sensilla campaniformia (SCa). These differences may vary according to their own biological habits. For research purposes, the observed difference in the sensillum distribution and function between the two fir longhorn beetles will greatly facilitate the design of better semiochemical control methods of these insect pests.

## Introduction

*Cunninghamia lanceolata* (Lambert) Hooker, Chinese fir, is one of the most commonly planted species in cultivated fast-growing timber forests of East and Southeast Asia, such as China, Japan, Laos, Vietnam, and neighboring countries, is popularly used for home and gardens due to its soft but durable wood that is easily workable [1]. Therefore, it has great

**Funding:** This research was funded by National Natural Science Foundation of China (31660626). The funders had no role in study design, data collection and analysis, decision to publish, or preparation of the manuscript.

**Competing interests:** The authors have declared that no competing interests exist.

economic and ornamental value. *Allotraeus asiaticus* Schwarzer and *Callidiellum villosulum* Fairmaire are two reported pest species of China fir (*C. lanceolata*) [2,3]. Indigenous to Southern China, there are few reports on the biology of *A. asiaticus* [4]. On the other hand, *C. villosulum* is only found in East Asia, but is occasionally intercepted in wood and wood products in US, Malta and Japan [2,5,6]. To avoid its proliferation in other regions of the world, it is necessary to systematically study the two beetles in order to plan preventive measures such as developing detection tools.

As phytophagous insects, Cerambycidae are economically important pests of street trees and forests [7], with more than 36,000 described species worldwide [8]. In recent years, more research on pheromones and behavior of longhorn beetles has been done, such as host location [7,9], mating [10,11], oviposition [12], thanatosis (death-feigning) [13], and sex and aggregation pheromones [14]. Behavior is inseparable from signaling and signal reception. In the insect world, signal perception and antennal structure are interrelated, and insects rely on these structures for gathering information on the environment and chemical communication with conspecifics and interspecifics [15].

Insect antennae have sensory receptors called sensilla. Several morphological structures are adapted to the primary function of insect antennae, such as omnidirectional movements and sensing adequate area. Antennae shaking during walking, the antennal sensilla can facilitate the insect to recognize different stimuli in the environment and are specialized for taste, olfaction, hygroreception, thermoreception or mechanoreception [16]. Because of functional requirements, different sensilla specialize in different morphological characteristics. Sensilla are differentiated according to morphology such as Böhm bristles, chaetica, trichodea, basiconca, coeloconica, campaniformia, and others. These sensilla types have been reported in a variety of insects [17–19]. Therefore, the distribution and number of sensilla with different functions may affect the reception of information, as related to specific searching behavior of insects [20].

The purpose of this study was to compare antennal morphology and sensilla ultrastructure between *A. asiaticus* and *C. villosulum* and between the sexes of each species via scanning electron microscopy (SEM) techniques. To our knowledge, there have been no studies on the description of the antennal sensilla of these two species using SEM techniques. The two species of beetle both belong to family Cerambycidae, and the main flight season is from March to May [21,22]. Thus, by comparing the species, numbers and distribution of antennal sensilla between the two species of stem-boring pest that utilize the same host, it will contribute to a better understanding of the differences in their living habits and behaviors. Meanwhile, this descriptive work will also provide the theoretical basis for future work on pheromone identification and development of prevention and control techniques for these two pests.

## Materials and methods

### Insects

The adults of *A. asiaticus* and *C. villosulum* (Fig 1) were captured in Chinese fir forest at Gaofeng Forest Park (22ptured in Chinese fir forest at rk on pheromone identification and development of prevention and control techniques for these tAlphaScents, Portland, OR) that were coated with a 50% aqueous dilution of the fluoropolymer Fluon to render traps more slippery and improve trap efficiency [3]. Traps were hung on a branch or on a wooden strut and kept 0.5–1 m high off the ground. Five male and five female adults of *A. asiaticus* and *C. villosulum* were selected from the trap catches, and were placed into a freezer at −20°C. After 30 minutes, the adults were removed and their antennae were excised under a stereomicroscope PX-1 (Camsonar Technology Co. Ltd, Beijing, China). The antennae were stored in a 75% alcohol

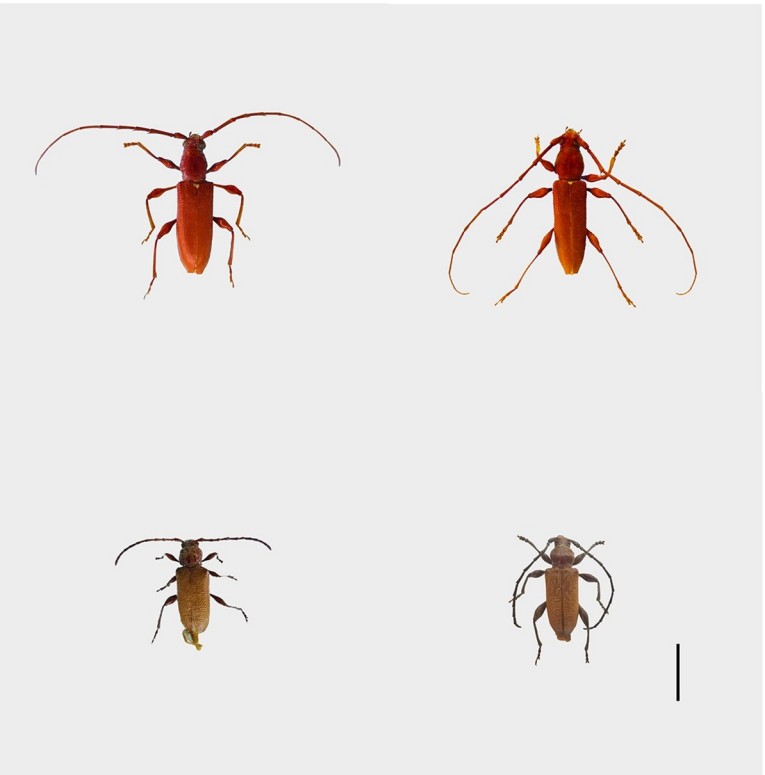

**Fig 1.** Adults of *A. asiaticus* (A, B) and *C. villosulum* (C, D). Scale bars: (A, B, C,D) = 5 mm.

solution until examination. All animal experiments were approved by the Institutional Animal Care and Use Committee of Guangxi University and animal care and use protocol are based upon the National Institutes of Health (NIH), USA.

## Scanning electron microscopy

The antennae were cleaned three times by distilled water in an ultrasonic bath JP-010T (Sky-men Cleaning Equipment CO., Ltd., Shenzhen, China) at 250 W for 360 seconds each, and were then fixed separately in 2.5% glutaraldehyde at 4°C for 12 h. The antennae were dehydrated through an ascending ethanol series of 75%, 80%, 85%, 90%, 95% and 100% at 10 min intervals. The prepared antennae were stored in a cleaned and dried glass petri dishes container that was air-dried for 12 h. After drying, the specimens were mounted on a holder using double-sided sticky tape and sputter coated with gold-palladium. Samples were put into a holder with double-sided adhesive tape (dorsal, ventral). Subsequently, the prepared samples were scanned by the electron microscope (model S-3400 N, Hitachi, Japan) at an accelerating voltage of 5–10 kV. Images were named respectively and stored on a computer.

## Dates collection and analysis

Identification and classification of the sensilla types and the terminology used in this work was based on studies of Schneider [21] and Zacharuk [22]. The length, and intermediate width of antenna and various sensilla were counted by the software Image J Launcher Version 1.44p (Broken Symmetry Soft-ware, National Institutes of Health, USA). According to the images of SEM, these sensillar distribution patterns were precisely described using the software Adobe

Photoshop Version CS6 and counted individually. The general morphological characteristics of antennae of *A. asiaticus* and *C. villosulum* were also drawn by Adobe Photoshop Version CS6. The quantity and distribution of each type of sensillum was analyzed between the antennae of both sexes of the two beetles. One-way ANOVA was applied to determine possible differences in antennal sensilla between the sexes of each species and the qualities of spine between different flagellomeres of the same sex of *A. asiaticus*, by using SPSS statistical software package version 25.0 (SPSS Inc., Chicago, IL, USA). In all, individual samples per type were subjected to a quantitative analysis in both side (dorsal, ventral) of antenna. A *t*-test or one-way ANOVA was applied to determine possible differentiation of the number, length, and width of antennal sensilla, and values were reported as mean ± SE (standard error). The significance level was set at 0.05. All graphs were made by SPSS 25.0 and GraphPad prism 6.01 (GraphPad Software, Inc., La Jolla, CA, USA).

## Results

### Gross morphology of antennae of *A. asiaticus* and *C. villosulum*

**Shape characteristics.**   Antennae of the two fir longhorn beetles both consist of three segments: scape (Sc), pedicel (Pd) and flagellum (nine flagellomere). There is a great difference between *A. asiaticus* and *C. villosulum* in the shape of the flagellum. The nine flagellomeres of *A. asiaticus* are all columnar (Figs 2A, 2E, 2G, 3C and 3D), but the 1st-6th subsegments of the flagellum of *A. asiaticus* appeared as a characteristic prism (Fig 2C and 2E). Besides, we also found that the 1st-5th subsegments of flagellum of *A. asiaticus* had a specialized spine on inner apex in both sexes (Table 1, Fig 3C and 3D). In *C. villosulum*, the structure of flagella was clear specialized as serrated shapes in 4th-8th subsegments of both sexes (Figs 2F, 3A and 3B). And the 9th flagella of male *C. villosulum* shown like the grain of *Poa supina* Schrad. (Figs 2H and 3B).

**Length and width of antennae.**   In *A. asiaticus*, there are no significant difference between males and females in the length of antennae (Table 2). In terms of width, female adult antennae were wider than male in the scape (t = 4.32, *p* = 0.00) and 3rd (t = 2.51, *p* = 0.03), 4th (t = 2.49, *p* = 0.03), 7th (t = 2.42, *p* = 0.04), 9th (t = 3.48, *p* = 0.00) subsegments of the flagellum (Table 2).

However, the antennae of male *C. villosulum* (11.25 ± 0.27 mm) were distinct significantly longer than female (7.04 ± 0.08 mm, t = -15.29, *p* = 0.00) (Table 2). Meanwhile, it was shown that the male adult antennae were wider than females in scape (t = -7.06, *p* = 0.00), pedicel (t = -3.15, *p* = 0.01) and 3rd (t = -3.16, *p* = 0.01) of flagellum (Table 2). On the contrary, female flagellomeres were wider than males in 7th (t = 2.18, *p* = 0.04) and 9th (t = 2.54, *p* = 0.02) subsegments of the flagella.

Comparing between different species, *A. asiaticus* were greater in antennal length than *C. villosulum* both in female (t = 13.19, *p* = 0.00) and male (t = 15.75, *p* = 0.00) (Table 2). In terms of width, antennae of flagellum both sexes of *A. asiaticus* were wider than *C. villosulum* from scape to 3rd subsegment of the flagella and female *A. asiaticus* (0.21 ± 0.01 mm) shown wider than female *C. villosulum* (0.18 ± 0.00 mm, t = 3.03, *p* = 0.13) in 4th flagella (Table 2). But it is interesting to note that both sexes of *C. villosulum* were tended to be wider than *A. asiaticus* in 6th (female: t = -2.64, *p* = 0.02; male: t = -2.55, *p* = 0.02), 7th (female: t = -4.03, *p* = 0.00; male: t = -6.88, *p* = 0.00), 8th (female: t = -3.92, *p* = 0.00; male: t = -5.15, *p* = 0.00), and 9th (female: t = -3.80, *p* = 0.00; male: t = -2.80, *p* = 0.02) subsegment of the flagella (Table 2).

### Types of antennal sensillum of *A. asiaticus* and *C. villosulum*

Four different morphological types of sensilla both were found on the antennae of *A. asiaticus* and *C. villosulum*: Böhm bristle (Bb), sensilla trichodea (ST I, II), sensilla basiconica (SB I, II,

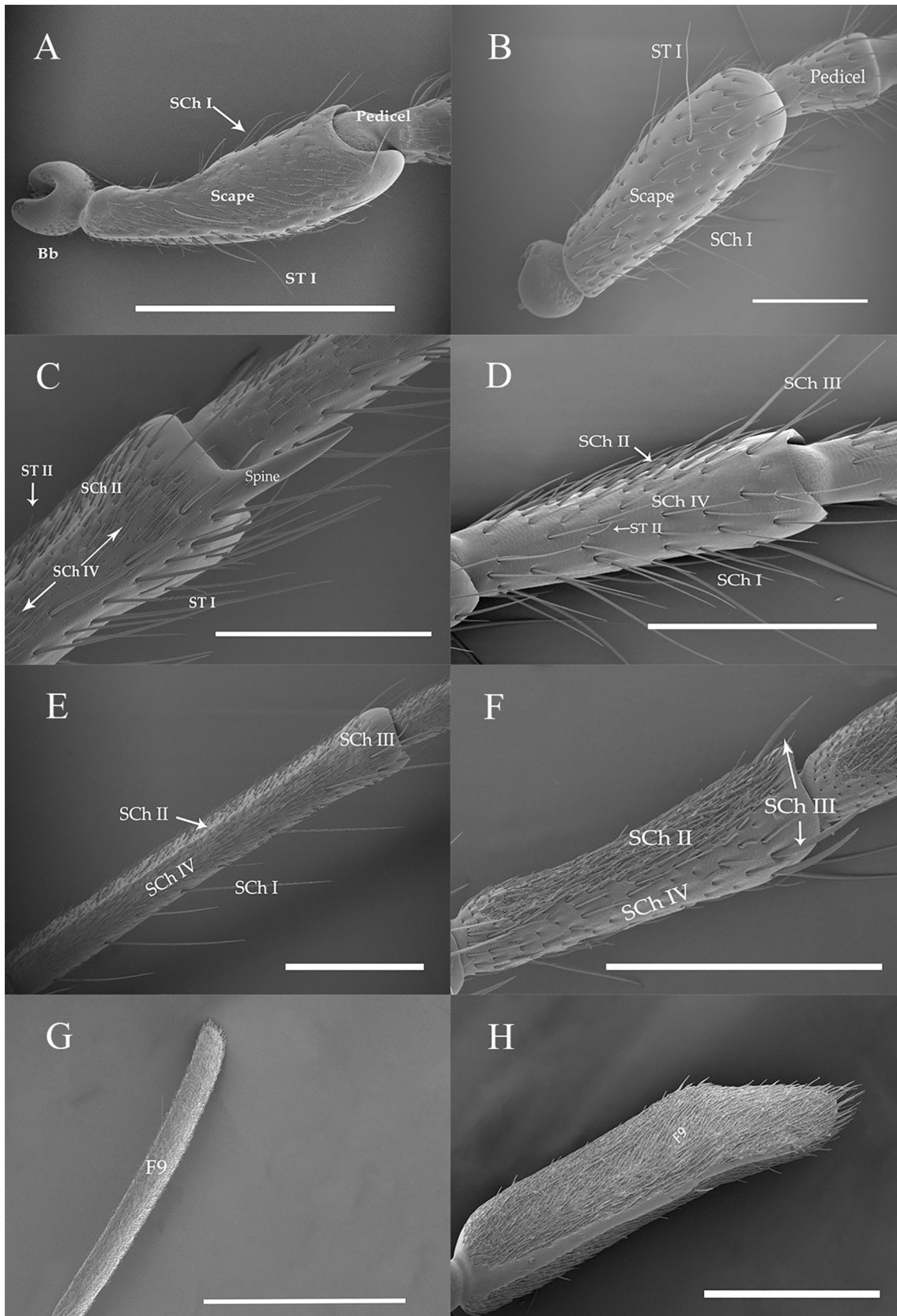

**Fig 2. Morphological characteristics of antennae of *A. asiaticus* and *C. villosulum*.** (A) Scape (Sc) and Pedicel (Pe) of male *A. asiaticus*. (B) Scape (Sc) and Pedicel (Pe) of male *C. villosulum*. (C) View of the 2nd flagellomere of female *A. asiaticus*. (D) View of the 2nd flagellomere of female *C. villosulum*. (E) View of the 4th flagellomere of female *A. asiaticus*. (F) View of the 4th flagellomere of female *C. villosulum*. (G) View of the 9th flagellomere of male *A. asiaticus*. (H) View of the 9th flagellomere of male *C. villosulum*. Scale bars: (A, E) = 1000 μm; (B, C, F, G, H) = 500 μm; (D) = 300 μm.

III), sensilla chaetica (SCh I, II, III, IV). One more morphological type of sensilla was found on the antennae of *C. villosulum* compared to *A. asiaticus*, which named sensilla campaniformia (SCa). There was no clear sexual dimorphism in the species and distribution of the antennal sensilla between different sexes of two fir longhorn beetles, respectively. However, there was significant difference in the number of the antennal sensilla between different sexes and species (Fig 4). The length, width and morphological characteristics of antennal sensilla are summarized in Table 3, respectively.

**Sensilla chaetica.**   Sensilla chaetica were widely distributed on antennae. It was spread over every segment of the antennae. The sensilla were more robust than other mechanical sensilla which could be further classified into four subtypes based on their morphological and ultra-micro characteristics: SCh I, II, III and IV. The number of SCh varies in the dorsal and ventral sides of the antennae of different segments of both sexes of *A. asiaticus* and *C. villosulum*, respectively (Fig 5).

Sensilla chaetica type I (SCh I) were widely distributed just on the ventral sides all over the antennae except the last segements of flagella (Table 3). Moreover, SCh I shown abundant on the ventral sides of 1st-3rd subsegments of the flagella of two fir beetles (Fig 5). What's more, it was the most robust sensillum in both fir beetle species. This kind of sensillum had a wide socket, slender tip and longitudinal grooves (Fig 6A and 6D). SCh I was the longest in female *A. asiaticus* with the mean length 450.50 ± 23.48 μm. However, the length in males is 343.11 ± 23.45 μm which was more robust than females of *C. villosulum* (Table 3).

Sensilla chaetica type II (SCh II) were the largest number of sensilum which looked slightly curved. It had a tight sockets, grooved wall and gradually tapers into a slender tip (Fig 6B, 6E and 6F). This type of sensillum were distributed differently between the two species. Particularly, there is a sexual difference in *C. villosulum* (Table 3). SCh II were distributed from scape to 9th subsegment of the flagella on the dorsal sides in *A. asiaticus*. Moreover, it was distributed on the ventral sides of 6th-9th subsegments of the flagellum in *A. asiaticus*. SChII were

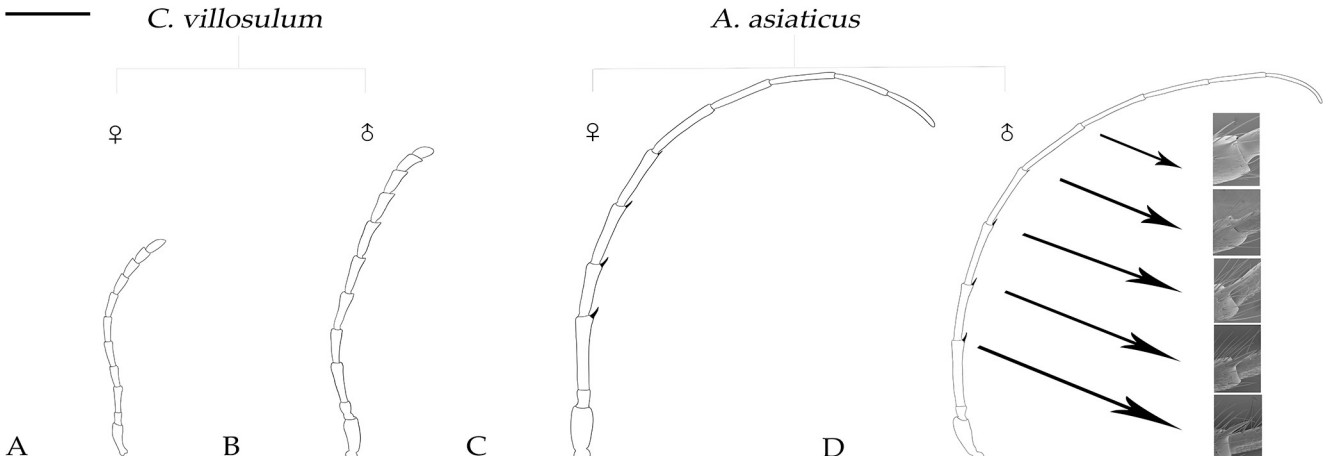

**Fig 3. General morphological characteristics of antennae of *A. asiaticus* and *C. villosulum*.** Scale bars just for the general morphological characteristics of antenna: (A, B, C, D) = 2 mm.

**Table 1. Length (μm) and width (μm) of spine of both sexes in *A. asiaticus*.**

| Sizes | Sex | Flagellomeres | | | | |
|---|---|---|---|---|---|---|
| | | F1 | F2 | F3 | F4 | F5 |
| Length | ♀ | 352.78 ± 18.35a * | 324.61 ± 22.19a | 246.20 ± 17.46b | 104.43 ± 14.00c * | 25.30 ± 3.42d |
| | ♂ | 276.57 ± 15.43a | 270.59 ± 14.46a | 234.36 ± 14.34a | 42.62 ± 2.11b | 46.05 ± 4.124b* |
| Width | ♀ | 98.04 ± 4.47a | 85.46 ± 3.76a | 72.55 ± 2.27b | 43.19 ± 2.07c * | 16.81 ± 0.89d |
| | ♂ | 85.84 ± 3.50a | 82.78 ± 4.17a | 72.19 ± 3.46a | 19.72 ± 0.75b | 15.71 ± 0.42b |

Note: Data were presented as mean ± SE, n = 5.

* indicate significant difference between male and female in length or width at 0.05 level using the *t*-test. Different letters in each row indicated significant difference at 0.05 level using Tukey's test.

only distributed from 3rd to 9th subsegments of the flagella on the dorsal side in *C. villosulum*. However, it was distributed differently between female (6th-9th subsegments of the flagella) and male (7th-9th subsegments of the flagella) on the ventral sides of *C. villosulum*. Over all, the length of this sensilla type in males was longer than females in both *A. asiaticus* and *C. villosulum* (Table 3).

Sensilla chaetica type III (SCh III) were visible at the junction of each segment of the flagellomeres (Fig 6A and 6D). This kind of sensillum has cuspidal tip, a wide socket and grooves on the surface. Moreover, SCh III appeared slightly curved. In distribution, there was no sexual differences in *A. asiaticus* (venter: 5th-8th flagella; dorsal: 1st-8th flagella) and *C. villosulum* (venter: 3rd-9th subsegments of the flagella; dorsal: 3rd-9th subsegments of the flagella), but there was a big difference between the venter and dorsal sides of its number both in two beetles (Table 3; Fig 5). The length of SCh III was longer in *A. asiaticus* (female: 263.43 ± 20.53 μm; male: 271.89 ± 23.28 μm) (Table 3). Meanwhile, there was no sexual differences in the length of the sensillum of *A. asiaticus* and *C. villosulum* (Table 3).

Sensilla chaetica type IV (SCh IV) were distributed from Sc to 6th flagella in both dorsal and ventral side of *C. villosulum* in both sexes (Table 3; Fig 5) with a tight socket, grooved wall

**Table 2. Length (mm) and width (mm) of antennal segments of both sexes in *A. asiaticus* and *C. villosulum*.**

| Antennomere | | Length | | | | Width | | | |
|---|---|---|---|---|---|---|---|---|---|
| | | *A. asiaticus* | | *C. villosulum* | | *A. asiaticus* | | *C. villosulum* | |
| | | ♀ | ♂ | ♀ | ♂ | ♀ | ♂ | ♀ | ♂ |
| Scape | | 1.56 ± 0.05 | 1.56 ± 0.02 | 0.88 ± 0.05 | 1.16 ± 0.12 * | 0.50 ± 0.02 * | 0.40 ± 0.02 | 0.28 ± 0.01 | 0.39 ± 0.01 * |
| Pedicel | | 0.45 ± 0.07 | 0.43 ± 0.03 | 0.28 ± 0.01 | 0.46 ± 0.02 * | 0.30 ± 0.01 | 0.29 ± 0.02 | 0.18 ± 0.00 | 0.22 ± 0.01 * |
| Flagellum | F1 | 2.16 ± 0.09 | 2.22 ± 0.07 | 0.75 ± 0.02 | 1.16 ± 0.08 * | 0.32 ± 0.01 | 0.28 ± 0.01 | 0.20 ± 0.01 | 0.21 ± 0.01 |
| | F2 | 1.63 ± 0.09 | 1.81 ± 0.05 | 0.63 ± 0.01 | 1.05 ± 0.03 * | 0.27 ± 0.01 | 0.26 ± 0.01 | 0.21 ± 0.01 | 0.22 ± 0.01 |
| | F3 | 1.90 ± 0.10 | 1.86 ± 0.08 | 0.77 ± 0.01 | 1.24 ± 0.05 * | 0.24 ± 0.01 * | 0.21 ± 0.01 | 0.18 ± 0.00 | 0.20 ± 0.01* |
| | F4 | 1.95 ± 0.10 | 1.92 ± 0.09 | 0.76 ± 0.01 | 1.18 ± 0.03 * | 0.21 ± 0.01 * | 0.18 ± 0.00 | 0.18 ± 0.00 | 0.19 ± 0.01 |
| | F5 | 1.98 ± 0.15 | 1.90 ± 0.04 | 0.70 ± 0.01 | 1.16 ± 0.09 * | 0.19 ± 0.01 | 0.17 ± 0.00 | 0.19 ± 0.01 | 0.19 ± 0.01 |
| | F6 | 1.74 ± 0.11 | 1.82 ± 0.09 | 0.59 ± 0.01 | 1.00 ± 0.02 * | 0.18 ± 0.01 | 0.17 ± 0.01 | 0.20 ± 0.01 | 0.19 ± 0.01 |
| | F7 | 1.74 ± 0.11 | 1.80 ± 0.08 | 0.56 ± 0.01 | 0.92 ± 0.02 * | 0.17 ± 0.01 * | 0.15 ± 0.00 | 0.20 ± 0.00 * | 0.19 ± 0.01 |
| | F8 | 1.61 ± 0.09 | 1.83 ± 0.06 | 0.49 ± 0.01 | 0.82 ± 0.03 * | 0.16 ± 0.01 | 0.15 ± 0.00 | 0.21 ± 0.01 | 0.19 ± 0.01 |
| | F9 | 1.91 ± 0.14 | 2.12 ± 0.08 | 0.63 ± 0.01 | 1.12 ± 0.04 * | 0.16 ± 0.00 * | 0.14 ± 0.00 | 0.20 ± 0.01 * | 0.17 ± 0.01 |
| Total | | 18.62 ± 0.87 | 19.65 ± 0.45 | 7.04 ± 0.08 | 11.25 ± 0.27 * | | | | |

Note: Data were presented as mean ± SE, n = 10.

* indicate significant difference in length or width at 0.05 level using the *t*-test between male and female of the same species.

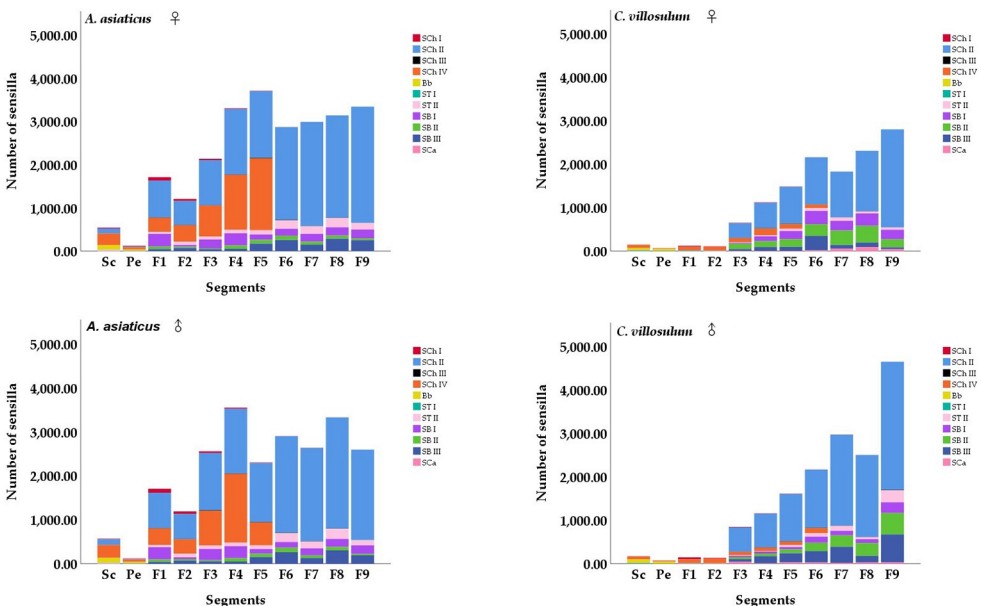

**Fig 4. Number and distribution of sensilla on the antennae of different segments of both sexes of *A. asiaticus* and *C. villosulum* (mean ± SE, n = 5).**

and gradually tapers to a slender tip (Fig 6B, 6C, 6D and 6F). Meanwhile, it was found in Sc-5th subsegment of the flagella on the dorsal side of *A. asiaticus* in both sexes. However, it had different distribution patterns in male (4th-5th subsegments of the flagella) and female (3rd-5th subsegments of the flagella) *A. asiaticus* (Table 3; Fig 5). Similarly, there was a big difference between the venter and dorsal sides of the quantity (Fig 5).The length of SCh IV was longer in *C. villosulum* (female: 97.86 ± 4.09 μm; male: 99.31 ± 9.23 μm) than *A. asiaticus* (female: 61.27 ± 2.10 μm; male: 66.63 ± 2.73 μm; Table 3).

**Böhm bristles.**   Böhm bristles (Bb) were observed on the base of the scape and pedicel, between the Head-Scape and Scape-Pedicel, respectively. However, no Bb were found on the ventral side of pedicel in *A. asiaticus*in both sexes (Table 3; Fig 7). Bb had a thorn-like structure with a wide socket. Besides, this type of sensillum had a smooth surface (Fig 8C and 8F). In female *A. asiaticus*, Bb had shown a salient advantage in length (47.01 ± 2.27 μm) and width (3.46 ± 0.11 μm) (Table 3). The number of Bb varies in the dorsal and ventral surface of the antennae of different segments of both sexes of *A. asiaticus* and *C. villosulum*, respectively (Fig 6).

## Sensilla trichodea

Sensilla trichodea were spread all over all segments of the antennae of *A. asiaticus* and *C. villosulum*. It had hair-like protruding receptors with grooves and perforated surfaces. The sensillum could been further classified into two subtypes based on their morphological and ultramicro-structural characteristics: ST I, II. The number of ST varies in the dorsal and ventral side of the antennae of different segments of both sexes of *A. asiaticus* and *C. villosulum*, respectively (Fig 7).

Sensilla trichodea type I (ST I) was present only on the dorsal side of the antennae of *A. asiaticus* (Sc-F9, F9 = the 9th flagellomere) and *C. villosulum* (Sc-3rd flagella) (Table 3; Fig 7). It inserted into a broadened socket and was long, slender, and curved with a grooved surface (Table 3; Fig 8A and 8B). ST I was relatively longer in male *C. villosulum* (417.80 ± 19.44 μm)

**Table 3. Morphological characteristics of the antennal sensilla of both sexes in *A. asiaticus* and *C. villosulum*.**

| Subtype | Species *A. asiaticus* L ♀ | *A. asiaticus* L ♂ | *A. asiaticus* W ♀ | *A. asiaticus* W ♂ | *C. villosulum* L ♀ | *C. villosulum* L ♂ | *C. villosulum* W ♀ | *C. villosulum* W ♂ | Side | Dist. *A. asiaticus* ♀ | Dist. *A. asiaticus* ♂ | Dist. *C. villosulum* ♀ | Dist. *C. villosulum* ♂ | Tip | Wall | Shape | Socket |
|---|---|---|---|---|---|---|---|---|---|---|---|---|---|---|---|---|---|
| SCh I | 450.50 ± 23.48* | 299.40 ± 15.82 | 9.95 ± 0.58 | 8.02 ± 0.26 | 234.09 ± 16.83 | 343.11 ± 23.45* | 8.40 ± 0.25 | 10.59 ± 3.7* | V | Sc-F8 | | Sc-F8 | | Slender | Grooved | Straight or Slightly curved | Wide |
| | | | | | | | | | D | — | | — | | | | | |
| SCh II | 53.34 ± 0.55 | 59.58 ± 0.55* | 4.25 ± 0.14 | 4.71 ± 0.19 | 40.94 ± 0.52 | 53.70 ± 1.28* | 3.86 ± 0.08 | 4.68 ± 0.11* | V | F6-F9 | | F6-F9 | F7-F9 | Slender | Grooved | Slightly curved | Tight |
| | | | | | | | | | D | Sc-F9 | | F3-F9 | | | | | |
| SCh III | 263.43 ± 20.53 | 271.89 ± 23.28 | 9.61 ± 0.40 | 10.46 ± 0.40 | 135.34 ± 10.38 | 180.00 ± 9.00* | 8.75 ± 0.56 | 12.78 ± 0.52* | V | F5-F8 | | F3-F9 | | Slender | Grooved | Slightly curved | Wide |
| | | | | | | | | | D | F1-F8 | | F3-F9 | | | | | |
| SCh IV | 61.27 ± 2.10 | 66.63 ± 2.73 | 3.12 ± 0.12 | 4.48 ± 0.20* | 97.86 ± 4.09 | 99.31 ± 9.23 | 4.34 ± 0.12 | 4.47 ± 0.27 | V | F3-F5 | F4-F5 | Sc-F6 | | Slender | Grooved | Straight | Tight |
| | | | | | | | | | D | Sc-F5 | | Sc-F6 | | | | | |
| Bb | 47.01 ± 2.27* | 25.73 ± 0.98 | 3.46 ± 0.11* | 2.66 ± 0.07 | 28.98 ± 2.02 | 34.10 ± 2.60 | 3.07 ± 0.12 | 3.38 ± 0.09* | V | Sc | | Sc, Pe | | Sharp | Smooth | Straight | Wide |
| | | | | | | | | | D | Sc, Pe | | Sc, Pe | | | | | |
| ST I | 377.39 ± 12.62 | 377.42 ± 16.18 | 12.26 ± 0.35 | 11.71 ± 0.50 | 321.32 ± 15.72 | 417.80 ± 19.44* | 8.89 ± 0.33 | 10.70 ± 0.43* | V | — | | — | | Slender | Grooved | Curved | Wide |
| | | | | | | | | | D | Sc-F9 | | Sc-F3 | | | | | |
| ST II | 54.31 ± 3.09 | 53.91 ± 3.37 | 4.08 ± 0.17 | 3.94 ± 0.18 | 47.11 ± 2.31 | 44.98 ± 2.67 | 3.68 ± 0.11 | 3.41 ± 0.19 | V | Sc-F9 | | F3-F9 | | Blunt | Grooved | Slightly curved | Wide |
| | | | | | | | | | D | Sc-F9 | | Sc-F9 | | | | | |
| SB I | 7.79 ± 0.25 | 8.28 ± 0.26 | 2.17 ± 0.07* | 1.92 ± 0.05 | 9.28 ± 0.54 | 9.45 ± 0.38 | 2.10 ± 0.09 | 1.93 ± 0.05 | V | F6-F9 | | F6-F9 | F7-F9 | Sharp | Smooth | Straight | Ridgy |
| | | | | | | | | | D | F1-F9 | | F3-F9 | | | | | |
| SB II | 8.96 ± 0.29 | 11.47 ± 0.38* | 2.49 ± 0.08* | 2.23 ± 0.08 | 11.81 ± 0.44 | 11.53 ± 0.27 | 2.31 ± 0.09* | 1.94 ± 0.08 | V | F6-F9 | | F6-F9 | F7-F9 | Blunt | Smooth | Straight | Ridgy |
| | | | | | | | | | D | F1-F9 | | F3-F9 | | | | | |
| SB III | 16.28 ± 0.90 | 16.26 ± 0.64 | 2.58 ± 0.13 | 2.36 ± 0.09 | 18.06 ± 1.12 | 19.66 ± 0.87 | 2.28 ± 0.12 | 2.08 ± 0.07 | V | F6-F9 | | F6-F9 | F7-F9 | Slender or Blunt | Smooth | Slightly curved | Ridgy |
| | | | | | | | | | D | F1-F9 | | F3-F9 | | | | | |
| Sca | — | — | — | — | 1.62 ± 0.01* | 1.59 ± 0.01 | 6.32 ± 0.17* | 5.70 ± 0.12 | V | — | | F6-F9 | F7-F9 | Blunt | Smooth | Campaniform | Ridgy & Tight |
| | | | | | | | | | D | — | | F1-F9 | | | | | |

Note: Data were presented as mean ± SE, n = 20.

* indicate significant difference between male and female of the same species at 0.05 level using the *t*-test. " ——— " indicated the sensilla was non-existent. L = Length, W = Width, D = Dorsal, V = Venter.

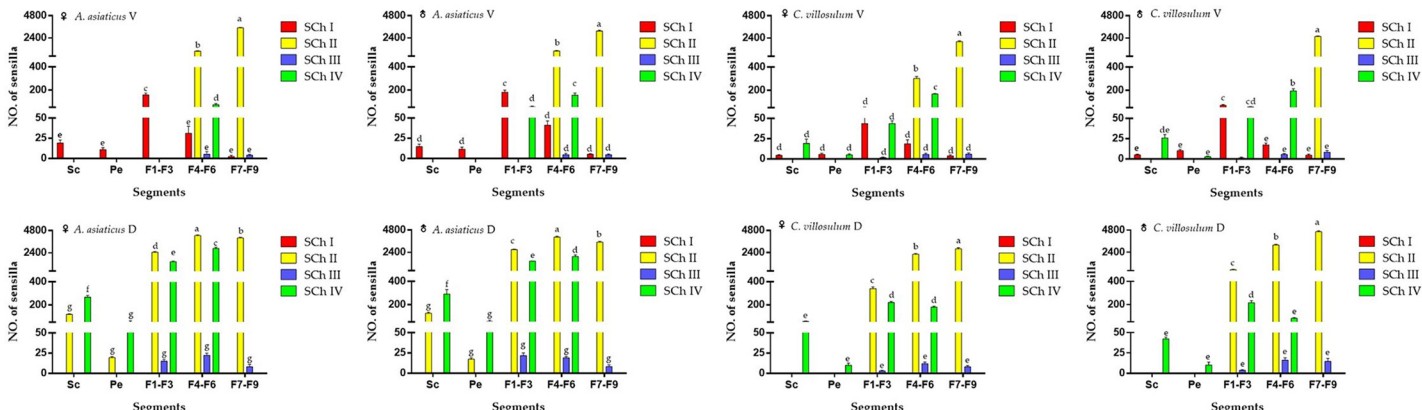

**Fig 5. Number and distribution of SCh (I, II, III, IV) on different segments of antennae in both sexes of *A. asiaticus* and *C. villosulum* (mean ± SE, n = 5).** D, dorsal side. V, ventral side. Different letters in each column indicated significant difference at 0.05 level using Tukey's test among sensillium on various segments of antennae.

than in both sexes of two beetles (Table 3). In contrast, ST I of female *C. villosulum* were the shortest (321.32 ± 15.72 μm) (Table 3).

Sensilla trichodea type II (ST II) were visible at the junction of flagellomeres (Fig 8D and 8E) which was curved hairs with blunt tips. Either the length or the width, there is no difference between the sexes in both beetles. ST II was distributed from Sc to 9th flagellomere on the dorsal side of antennae of *A. asiaticus* and *C. villosulum*. Where as, it shown differences between *A. asiaticus* (Sc-9th flagellomere) and *C. villosulum* (Sc-3rd flagellomere) on the ventral side. In total number, male *A. asiaticus* exhibited large quantities in 1st-3rd subsegments of the flagella while females had high abundance distributed in the 7th-9th subsegments of the flagella (Table 4). On the other hand, sexes of *C. villosulum* mainly showed differences in 7th-9th subsegments of the flagella that female had high abundance on ventral side with male shown an inverse distribution.

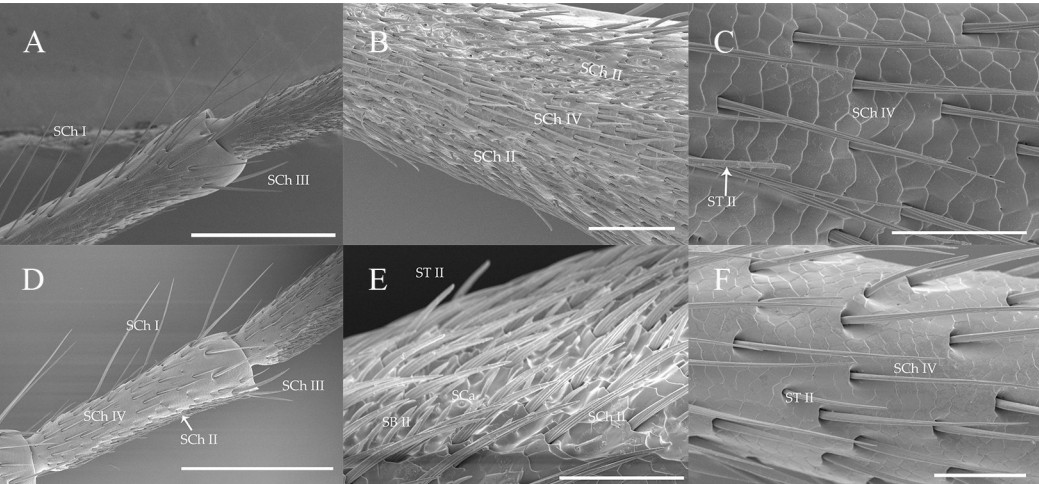

**Fig 6. The morphology of sensilla chaetica on the antennae of *A. asiaticus* and *C. villosulum*.** (A) SCh I and SCh III of *A. asiaticus*. (B) SCh II and SCh IV of *A. asiaticus*. (C) SCh IV of *A. asiaticus*. (D) SCh I and SCh III of *C. villosulum*. (E) SCh II of *C. villosulum*. (F) SCh IV of *C. villosulum*. Scale bars: (D) = 500 μm; (A) = 400 μm;(B) = 100 μm;(E, F) = 50 μm;(C) = 40 μm.

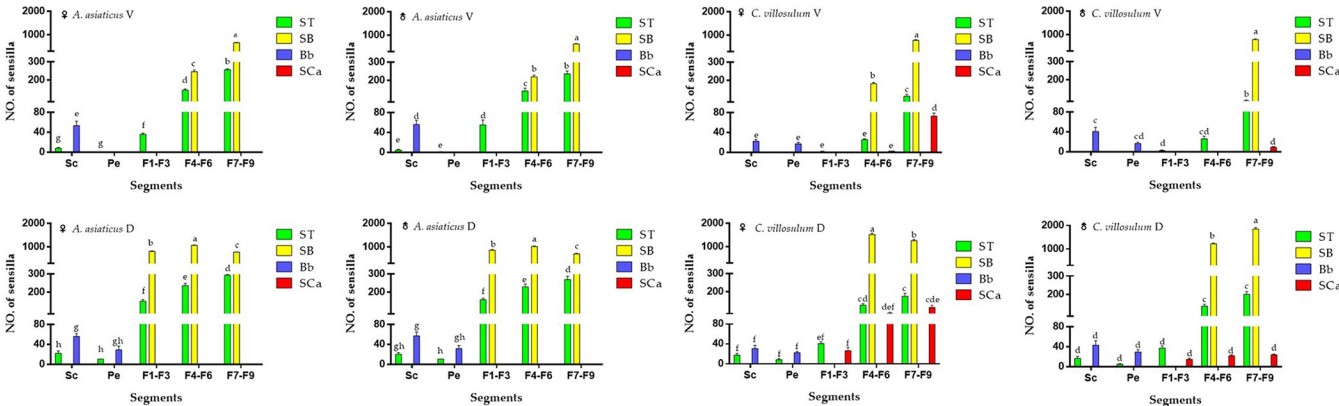

**Fig 7. Number and distribution of ST, SB, Bb, SCa on different segments of antennae in both sexes of *A. asiaticus* and *C. villosulum* (mean ± SE, n = 5).** D, dorsal side. V, ventral side. Different letters in each column indicated significant difference at 0.05 level using Tukey's test among sensillium on various segments of antennae.

**Sensilla campaniformia.** Sensilla campaniformia (SCa) were only found on the antennae of *C. villosulum* which were distributed from 1st-9th subsegments of the flagella on the dorsal side (Fig 7). However, the distribution on the ventral surface was different between males (7th-9th flagella) and females (6th-9th flagellomeres) (Table 3). It is a dome-shaped sensilla whose cuticular collar surrounding the central small cap forms a slightly raised dome (Fig 8G, 8H and 8I). This sensilla on the female *C. villosulum* (L: 1.62 ± 0.01 μm, W: 6.32 ± 0.17 μm) were more pronounced than males (L: 1.59 ± 0.01 μm, W: 5.70 ± 0.12 μm) (Table 3).

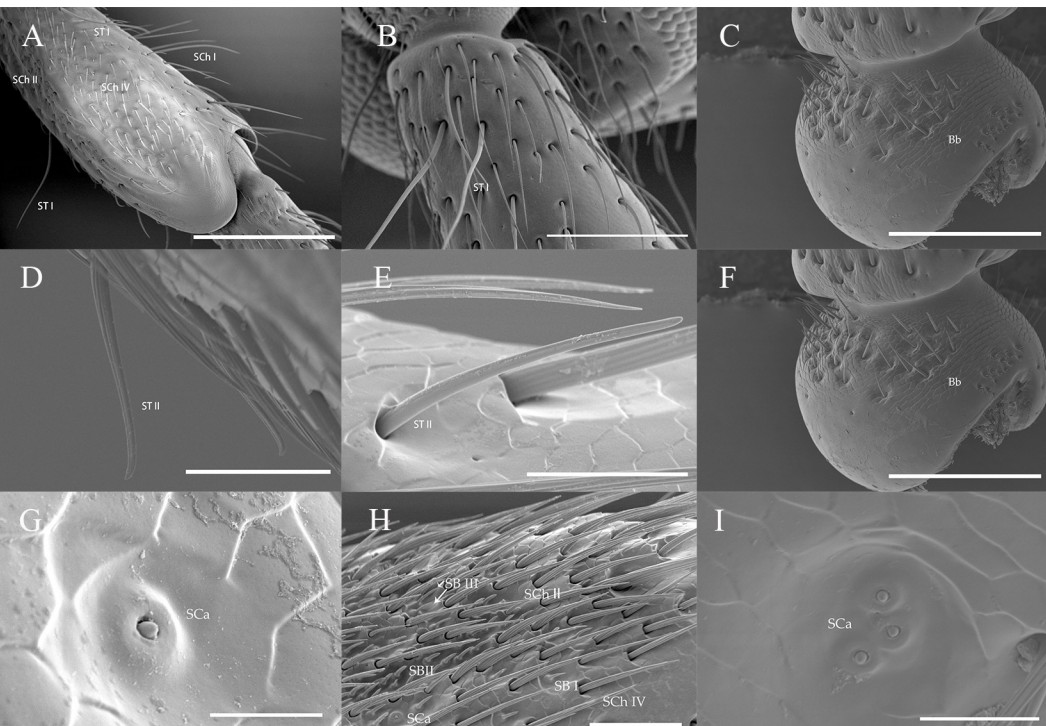

**Fig 8. The morphology of ST, Bb and SCa on antennae of *A. asiaticus* and *C. villosulum*.** (A) ST I of *A. asiaticus*. (B) ST I of *C. villosulum*. (C) T Bb of *C. villosulum*. (D) ST II of *A. asiaticus*. (E) ST II of *C. villosulum*. (F) Bb of *A. asiaticus*. (G, H, I) SCa of *C. villosulum*. Scale bars: (A) = 500 μm; (B, C) = 200 μm; (F, H) = 50 μm; (D) = 40 μm; (E) = 30 μm; (G) = 20 μm.

**Table 4. Number and distribution of ST and SB in the dorsal and ventral surface of different antennae segments in both sexes of *A. asiaticus* and *C. villosulum*.**

| Subtype | Species | Sex | Segments | | | | | | | | | | |
| --- | --- | --- | --- | --- | --- | --- | --- | --- | --- | --- | --- | --- |
| | | | Scape | | Pedicel | | F1-F3 | | F4-F6 | | F7-F9 | |
| | | | Venter | Dorsal | Venter | Dorsal | Venter | Dorsal | Venter | Dorsal | Venter | Dorsal |
| ST I | *A. asiaticus* | ♀ | — | 2.80 ± 0.58 | — | 2.40 ± 0.40 | — | 9.00 ± 0.95 | — | 1.40 ± 0.40 | — | 1.40 ± 0.40 |
| | | ♂ | — | 4.60 ± 0.40* | — | 3.40 ± 0.24 | — | 6.80 ± 0.58 | — | 1.80 ± 0.73 | — | 1.20 ± 0.49 |
| | *C. villosulum* | ♀ | — | 11.2 ± 0.86 | — | 3.80 ± 0.37 | — | 11.40 ± 0.98 | — | — | — | — |
| | | ♂ | — | 12.2 ± 1.69 | — | 3.00 ± 0.45 | — | 10.20 ± 1.11 | — | — | — | — |
| ST II | *A. asiaticus* | ♀ | 8.00 ± 0.84* | 19.40 ± 2.42 | 0.60 ± 0.40 | 8.60 ± 1.08 | 36.00 ± 1.14 | 141.20 ± 3.23 | 144.20 ± 3.06 | 235.00 ± 5.29 | 257.00 ± 1.70* | 292.00 ± 1.38* |
| | | ♂ | 4.60 ± 0.75 | 16.00 ± 1.14 | 1.00 ± 0.32 | 10.00 ± 1.05 | 55.40 ± 4.41* | 152.20 ± 3.65* | 141.80 ± 6.64 | 227.20 ± 7.11 | 236.40 ± 6.64 | 269.00 ± 7.34 |
| | *C. villosulum* | ♀ | — | 6.2 ± 0.66 | — | 4.40 ± 0.68* | 1.80 ± 0.37 | 29.20 ± 1.98 | 25.80 ± 0.8b | 125.40 ± 4.20 | 112.60 ± 4.50* | 175.40 ± 7.42 |
| | | ♂ | — | 4.40 ± 0.51 | — | 1.80 ± 0.37 | 2.80 ± 0.37 | 27.20 ± 1.20 | 26.00 ± 2.39 | 135.60 ± 4.45 | 82.80 ± 2.54 | 200.40 ± 6.99* |
| SB I | *A. asiaticus* | ♀ | — | — | — | — | — | 534.80 ± 13.90 | 88.40 ± 3.22* | 474.00 ± 10.39 | 256.80 ± 1.77 | 319.80 ± 1.96* |
| | | ♂ | — | — | — | — | — | 568.60 ± 16.62* | 58.20 ± 2.62 | 451.80 ± 12.72 | 251.00 ± 7.91 | 283.40 ± 8.47 |
| | *C. villosulum* | ♀ | — | — | — | — | — | 23.20 ± 2.60 | 60.80 ± 2.08* | 539.60 ± 19.77* | 254.20 ± 10.1* | 411.00 ± 17.30* |
| | | ♂ | — | — | — | — | — | 28.60 ± 2.73 | — | 240.20 ± 7.94 | 139.60 ± 4.30 | 318.80 ± 11.25 |
| SB II | *A. asiaticus* | ♀ | — | — | — | — | — | 114.60 ± 2.94 | 44.00 ± 1.61 | 223.60 ± 4.20 | 85.40 ± 0.81 | 97.60 ± 0.60* |
| | | ♂ | — | — | — | — | — | 118.00 ± 3.92 | 45.00 ± 2.05 | 212.60 ± 5.88 | 79.00 ± 2.49 | 90.20 ± 2.65 |
| | *C. villosulum* | ♀ | — | — | — | — | — | 123.60 ± 13.18* | 54.40 ± 1.89* | 532.40 ± 22.04* | 385.00 ± 15.65* | 629.40 ± 26.99 |
| | | ♂ | — | — | — | — | — | 39.00 ± 3.89 | — | 337.80 ± 11.26 | 276.20 ± 8.71 | 653.20 ± 23.46 |
| SB III | *A. asiaticus* | ♀ | — | — | — | — | — | 164.20 ± 3.76 | 113.40 ± 4.13 | 371.80 ± 7.87 | 325.20 ± 2.85* | 366.00 ± 8.76* |
| | | ♂ | — | — | — | — | — | 174.20 ± 4.68 | 116.20 ± 5.20 | 351.60 ± 10.00 | 300.00 ± 7.80 | 335.60 ± 8.76 |
| | *C. villosulum* | ♀ | — | — | — | — | — | 38.80 ± 4.16 | 67.00 ± 2.39* | 448.20 ± 17.03 | 140.80 ± 5.66 | 215.20 ± 8.69 |
| | | ♂ | — | — | — | — | — | 57.00 ± 5.64* | — | 644.00 ± 22.20* | 376.80 ± 11.74* | 879.80 ± 58.99* |

Note: Data were presented as mean ± SE, n = 5. "——" indicated the sensilla was non-existent.

* indicate significant difference between male and female of the same species at 0.05 level using the *t*-test.

**Sensilla basiconic.** Sensilla basiconic were interspersed on the flagellomeres (Fig 7). The distribution of sensilla basiconic on the two fir beetles shown that the dorsal side was obviously stronger than that of the ventral side (Table 4; Fig 7). In *A. asiaticus*, sensilla basiconic scattered over 1st-9th subsegments of the flagella on the dorsal side while it distributed in 6th-9th subsegments on ventral side (Table 3). Meanwhile in *C. villosulum*, sensilla basiconic were found in 3rd-9th subsegments of the flagella on the dorsal side while it distributed dimorphism on ventral side in female (6th-9th flagellomeres) and female (7th-9th flagella). On the other hand, it could be characterized as a cone-shaped prominence with a pedestal shape or a conically uplifted base. In the center of the base, there are small cone-shaped receptors with different shapes, which has chemical sensory functions such as smell perception and taste perception. The basiconic sensilla could be further classified into three types based on their surface micro-morphology: Sensilla SB I, II and III.

Sensilla basiconic type I (SB I) were cone-shaped and straight with a slightly pointed tip (Fig 9A and 9D) and a smooth surface. The length of SB I in *C. villosulum* (female: 9.28 ± 0.54 μm; male: 9.45 ± 0.38 μm) was longer than *A. asiaticus* (female: 7.79 ± 0.25 μm; male: 8.28 ± 0.26 μm) both without sexual differences (Table 3). In terms of distribution, female *C. villosulum* with plentiful SB I in 4th- 9th subsegments of the flagella on the both sides, and more than males overall (Table 4).

Sensilla basiconic type II (SB II) has no basic sockets and gradually tapers into the blunt tip with a smooth surface and only distributed on the flagellum (Table 3; Fig 9B and 9E). The length of this sensbnhmj,kjmnbnm,nbvkiuhnbvillan in male *A. asiaticus* (11.47 ± 0.38 μm) was longer than female (8.96 ± 0.44 μm, t = -5.22, p = 0.00). However, there was no sexual dimorphism in *C. villosulum* (female: 11.81 ± 0.44 μm, male: 11.53 ± 0.27 μm) (Table 3). Female *C. villosulum* also shown more than males overall in 4th- 9th subsegments of the flagella. In addition, the number of SB II in female *A. asiaticus* (97.60 ± 0.60) were significantly different compared to male (90.20 ± 2.65, t = 2.72, p < 0.05) on the 9th flagellomere (Table 4).

Sensilla basiconic type III (SB III) had a smooth surface and ridgy socket with a slender tip in *C. villosulum* (Fig 9F). However, its tip appears blunt in *A. asiaticus* (Fig 9C). In addition, SB III appears longer on the antennae of *C. villosulum* (female: 18.06 ± 1.12 μm, male: 19.66 ±0.87 μm) than *A. asiaticus* (female: 16.28 ± 0.90 μm, male: 16.26 ± 0.64 μm) both without

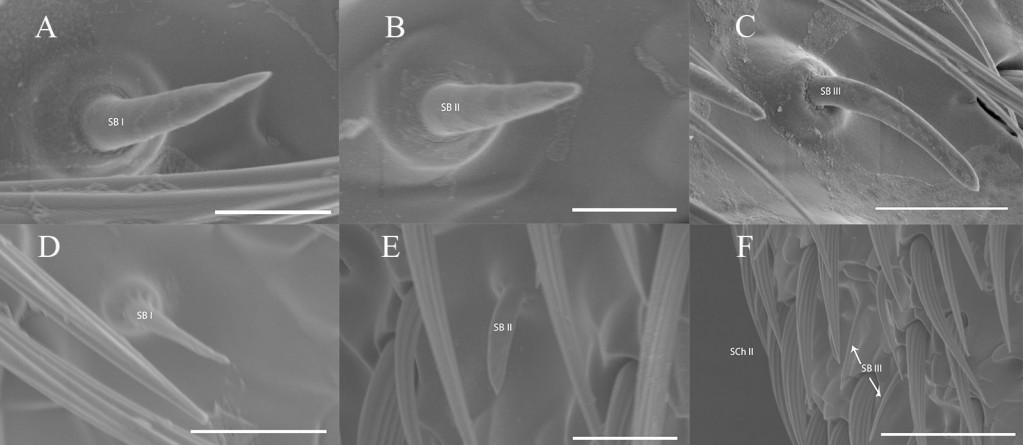

**Fig 9. High-resolution images for types of SB on the antennae of *A. asiaticus* and *C. villosulum*.** (A) The morphology of SB I of *A. asiaticus*. (B) The morphology of SB II of *A. asiaticus*. (C) The morphology of SB III of *A. asiaticus*. (D) The morphology of SB I of *C. villosulum*. (E) The morphology of SB II of *C. villosulum*. (F) The morphology of SB III of *C. villosulum*. Scale bars: (F) = 30 μm; (C, D, E) = 10 μm; (A, B) = 5 μm.

sexual differences (Table 3). In terms of distribution, SB III abundantly covered on flagellum in male *C. villosulum* compared to female (Table 4). But in *A. asiaticus*, numbers were only found significant difference between male and female in 7th-9th subsegments of the flagella on both sides (Table 4).

## Discussion

In the past half century, the antennal scanning electron microscopy of insects has been well-researched [23–25]. As phytophagous insects, Cerambycidae are economically important pests of forest, street trees and fruit trees [34]. Therefore, antenna scanning electron microscopy of cerambycid beetles has gained much recent research attention [26–28]. At present, the study of antennae ultrastructure of longicorn beetles is focused on the comparison between male and female adults of each species. However, the interspecific comparison of antennae ultra-structure of different longicorn beetles with the same host plants is very rare. Thus, we studied the ultrastructure of the antennae of *A. asiaticus* and *C. villosulum*. From this study, we found several significant differences between the two fir longhorn beetles in the antennal shape. The five flagellomeres of *C. villosulum* were specialized as serrated shapes in 4th-8th subsegments of the flagella with abundant olfactory sensillum. Those characters can greatly increase the effective sensing area of the antennae, enhancing the inductive force [29]. On the other hand, the cross section of *A. asiaticus* in 1st-6th subsegments of the flagella were pentagonal (Fig 1E). The sensilla distribution on each side was quite different. These structures are all related to olfactory traits which help to specialize local behaviors.

The morphological type and characteristics of sensillum on the antennae of *A. asiaticus* and *C. villosulum* are basically the same, but varies greatly with the quantity of each type. According to the external shape, dimensions and location, four subtypes of chaetica, three subtypes of basiconic, two subtypes of trichodea, one type of Böhm bristles were distinguished in both male and female adults of *A. asiaticus* and *C. villosulum*. Meanwhile, there is an extra sensilla named sensilla campaniformia in both sexes of *C. villosulum*. The functional and morphological characteristics of these five types of sensilla have been proven analogous in other beetles [26–28]. Sensilla campaniformia, as a kind of temperature and humidity sensor, is common in some kinds of insect, especially in the oral appendages [30–32], but rarely found on antennae of longhorn beetles [26]. This is related to the wood moisture and may be related to strict requirements of oviposition preference [33].

The main reported functions for sensilla chaetica and Böhm bristles show a mechanical function [26,34,35]. Meanwhile, some research also have shown that several subtypes of sensilla chaetica have an olfactory function [34], such as SCh II & IV in *A. asiaticus* and *C. villosulum*. Sensilla trichodea and sensilla basiconic, as two common olfactory sensors, have been found in a large number on the antennae of longhorn beetle [27,28]. According to these studies, the possible function of these sensillum was discussed on the basis of the morphological description and distribution. Only from the point of view of quantity, *A. asiaticus* outperforms *C. villosulum* (Fig 4), and the olfactory sensilla of *C. villosulum* has the advantage of quantity in the flagellomeres. The number of sensillum on the dorsal side of *A. asiaticus* and *C. villosulum* was larger than that on the ventral side. Moreover, the presence of olfactory sensillum was only found on the ventral side of last sections of distal part of the flagellum.

Mechanical sensillum in *A. asiaticus* and *C. villosulum*, consist of SCh I &III and Bb. Among them, SCh I is more robust than other mechanical sensilla which was only distributed on the ventral side of the antennae that are related to the mechanical induction function while landing (Table 3; Fig 5). In addition, we found males had higher abundance of this type compared to females on 4th-9th subsegments of the flagella in *A. asiaticus* (Table 3; Fig 5).

Meanwhile, the males were more developed than females both in measurements and quantities of *C. villosulum* (Table 3; Fig 5). In addition to landing on the host, the mating behavior of males requires holding the females, which may explain this morphology. SCh III are visible at the junction of some segments of the flagellomeres in *A. asiaticus* and *C. villosulum* are related to mechanical induction when the antennae are waving (Table 3, Fig 6A and 6D). Bb are a special type of sensilla chaetica located only on the joint region between the ommateum and the scape, as well as on the joint region between the scape and the pedicel. As the antennae are more developed, the mechanical sensillum (SCh III and Bb) of joints also show more prominence in *A. asiaticus* (Fig 7).

Olfactory sensillum, as a general trend, contains richer sensilla subtypes in *A. asiaticus* and *C. aillosulum*, including SCh (II, IV), ST (I, II), SB (I, II, III). Although they have the same subtypes of olfactory sensilla, there are some differences in quantity, length and width between the sensillum (Table 4; Figs 5 and 7). Previous study has shown that there were differences in aggregation-sex pheromone components between *C. villosulum* and *A. asiaticus* [3]. The results of field trials show that *C. villosulum* shown specific attraction to the blend of 3-hydroxyhexan-2-one and the pyrrole, while *A. asiaticus* was only specifically attracted to the pyrrole as a single component [36]. Furthermore, the results of this study were compared within species, and some other interesting conclusions were found (Fig 10). In *A. asiaticus*, there is no difference in the number of males and females caught during the field experiment, which is consistent with the small difference in the number, measurement and distribution of sensillum between males and females (Tables 3 and 4; Figs 4, 5, 7 and 10). On the contrary, there is a big difference between males and females in *C. villosulum* (Tables 3 and 4; Figs 4, 5 and 7), which supported by the field experimental result (Fig 10). Meanwhile, there was no significant difference in the number of olfactory sensilla between male and female as a whole in *A. asiaticus* (Fig 2). But SCh II, SB II has shown that the male sensilla of these types were longer in shape than in the female (Table 3). This result suggests that these two sensilla very likely involved in the identification of sex pheromone components. In *C. villosulum*, SB I & II were more numerous in females compared to males (Table 4), which may be related to the induction of a compound that is analogous to the oviposition pheromone. In contract, the male *C. villosulum* showed a particularly well-developed SCh IV and SB III (Fig 6), which may be related to the identification of participatory pheromones.

In examining the temperature and humidity sensing structures, *A. asiaticus* and *C. villosulum* were extremely different in structure in this regard. In *C. villosulum*, common temperature and humidity sensilla were spread all over their flagellomeres (Figs 4 and 7). This was

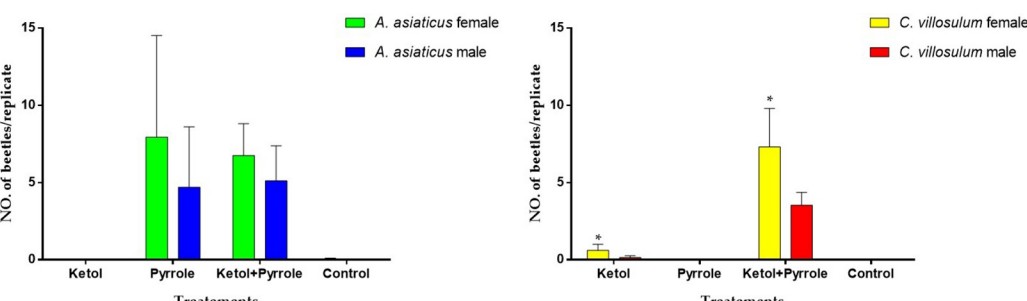

**Fig 10. Mean (± SE) numbers per replicate of adult beetles of different genders of *A. asiaticus* and *C. villosulum* that were caught during a field experiment conducted in Guangxi Autonomous Region, China, during March to April 2016.** Compound abbreviations: ketol—racemic 3-hydroxyhexan-2-one, pyrrole—1-(1H-pyrrol-2-yl)-1, 2-propanedione. * indicate significant difference between different objects of the same treatment at 0.05 level using the *t*-test.

related to the habit of laying eggs only on dry and injured fir branches [33]. Interestingly, F1-F5 flagellomeres of *A. asiaticus* had a specialized spine on the top of outer side in both sexes. There is no similar report in other longhorn beetle species. It has been reported that spines are found in other insects, for example styloconica that function as thermo-hygroreceptors [37]. So we speculate that this possibly plays a role as temperature and humidity sensing structures, however its exact functions need to be explored in future experiments.

This study aims to identify and characterize the external morphology and distribution of antennal sensillum types of *A. asiaticus* and *C. villosulum* using a SEM, and to compare the differences between species and sexes of these two fir longhorn beetles. Combining the research results of their living habits and other insects (especially other longhorn beetle species) with the results of this study, will provide valuable insights into possible functions of each sensillum. These results provide the necessary background information for future studies on the chemical ecology of these economically important longhorn beetle species. The observed difference in the sensilla distribution and function will greatly facilitate the design of better semiochemical control methods, for example more effective lures for survey and detection, for these pest insects.

## Conclusions

In this research, we found that the antennae of *C. villosulum* are sexual dimorphism, but *A. asiaticus* is opposite. Coincidentally, the conclusions in our past field experiment can reasonably prove this result. Moreover, there's a big difference between *A. asiaticus* and *C. villosulum* on antennae in terms of interspecies. First of all, the morphological characteristics of their antennae are quite different which related to their specific and unknown living habits and behaviors. This needs further study. Secondly, common temperature and humidity sensing sensilla were spread all over flagellomeres in *C. villosulum* which was related to the habit of laying eggs only on dry and injured fir branches. Last, the number of sensillum in the same types varied greatly among species. The result can well explain that the two longicorn beetles responded to different pheromones in our field experiment. These results merit further study with a particular focus on the chemosensory characteristics, which will greatly facilitate the design of better semiochemical control methods for these two fir longhorn beetles.

## Supporting information

**S1 File. Supplement tables.**
(XLSX)

## Acknowledgments

We are grateful to editors and two anonymous reviewers for helpful suggestions to improve the manuscript.

## Author Contributions

**Conceptualization:** Zishu Dong, Fugen Dou, Yubin Yang, Yujing Zhang, Zongyou Huang, Xialin Zheng, Xiaoyun Wang, Wen Lu.

**Data curation:** Zishu Dong, Wen Lu.

**Formal analysis:** Zishu Dong, Fugen Dou, Yubin Yang, Wen Lu.

**Investigation:** Zishu Dong, Jacob D. Wickham, Rong Tang, Xialin Zheng, Wen Lu.

**Methodology:** Zishu Dong, Yujing Zhang, Zongyou Huang.

**Resources:** Zishu Dong, Jacob D. Wickham, Rong Tang, Wen Lu.

**Software:** Zishu Dong, Yujing Zhang, Zongyou Huang, Xiaoyun Wang.

**Supervision:** Zishu Dong, Wen Lu.

**Validation:** Zishu Dong.

**Visualization:** Zishu Dong, Wen Lu.

**Writing – original draft:** Zishu Dong.

**Writing – review & editing:** Zishu Dong, Fugen Dou, Yubin Yang, Jacob D. Wickham, Xialin Zheng, Xiaoyun Wang, Wen Lu.

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
