## [Decision Letter · Decision Letter 0]

26 Aug 2020

PONE-D-20-25310

First description and comparison of the morphological and ultramicro characteristics of the antennal sensilla of two fir longhorn beetles

PLOS ONE

Dear Dr. Lu,

Thank you for submitting your manuscript to PLOS ONE. After careful consideration, we feel that it has merit but does not fully meet PLOS ONE’s publication criteria as it currently stands. Therefore, we invite you to submit a revised version of the manuscript that addresses the points raised during the review process.

We look forward to receiving your revised manuscript.

Kind regards,

Yulin Gao

Academic Editor

PLOS ONE

2. In your Methods section, please provide additional information regarding the permits you obtained for the work. Please ensure you have included the full name of the authority that approved the collection site access and, if no permits were required, a brief statement explaining why.

3. Please ensure that you include a title page within your main document. We do appreciate that you have a title page document uploaded as a separate file, however, as per our author guidelines (http://journals.plos.org/plosone/s/submission-guidelines#loc-title-page) we do require this to be part of the manuscript file itself and not uploaded separately.

"This research was funded by National Natural Science Foundation of China (31660626). The funders had no role in study design, data collection and analysis, decision to publish, or preparation of the manuscript. "

"This work was also supported financially by degree construction - funds for doctoral study abroad of Guangxi University in 2019."

Review Comments to the Author

Reviewer #1: The study proposed by Done et. al is based on antennal morphology analyses of A. asiaticus and C. villosulum using scanning electron microscopy based approach. The study described in this manuscript was conducted in China and it reports the first comprehensive list of antennal sensilla of two fir longhorn beetles, important pests in this region of the world. The overall objective of the study was accomplished as stated. The new information reported in the study builds upon what is already known about the antennal sensilla of other insects. The comparative analyses also validate some of the previously published information on morphological and ultramicro of antennal sensilla in other insect pests. Information from this study, can provide a better understanding of the molecular basis of olfaction and other chemical cues that regulate behavior in A. asiaticus and C. villosulum. This in turn, can provide a basis for developing alternative and better methods of controlling this pest.

This study is the first of its type on A. asiaticus and C. villosulum and it provides useful information on the comparative morphological and ultramicro analyses on the economically important insect pests. I believe it will contribute to the body of knowledge that is available on A. asiaticus and C. villosulum. In this regard I will recommend the manuscript for acceptance for publication subject to the comments addressed here.

1. As a whole，this article is tedious and repeated, such as the abstract, which should be compressed to more describe morphological and ultramicro characteristics.

2. There are many logical problems in this paper, such as “the study on the comparison between…” (lines 316-319), “during field trials, C. villosulum shown…” (lines 368-371) and “this is related to the identification of…” (lines 378-383), move logically from one idea to the nextlogically from one idea to the next and don't skip steps, revise it again.

3. Overall, the authors paid attention to details in writing of the manuscript.

Reviewer #2: In this study, the authors compared the antennal morphology and sensilla ultrastructure between A. asiaticus and C. villosulum and between the sexes of each species via scanning electron microscopy (SEM) techniques. The findings of the current study can contribute to a better understanding of the differences in their living habits and behaviors. Meanwhile, this descriptive work will also provide the theoretical basis for future work on pheromone identification and development of prevention and control techniques for these two pests. Overall, this study is interesting, the methods used are standard, the manuscript is well written scientifically and the data is sufficient to support its main conclusion. However, I have a few comments for improvement in my view.

Some suggestions:

Lines 15-17: Please rewrite this sentence more clearly.

Lines 21-22: I think no need of this sentence.

Lines 26-29: The given reference focus Chinese firs only in Fujian Province, China. Please provide references for their distribution in other countries or revised the statement which focus only China.

Lines 31: Replace indigenous by Indigenous..

Lines 43-45: Please provide references for these sentences.

Lines 108: Please use full scientific name at the start of a sentence. Check whole MS and correct it.

Lines 110-114: The use of three adversative conjunction ‘Interestingly’, ‘Meanwhile’ and ‘Nevertheless’ led to a logic miss. Please consider to rewrite these three sentences. Also, please check whole MS regarding this kind of mistakes.

Lines 131-132: P should be italic. Correct this in whole MS.

Lines 142-143: A. asiaticus and C. villosulum, should be italic.

Lines 158: Remove “a large” from sentence.

Lines 159: Replace sex by “sexes”.

Lines 177; 297-299; 378-379: The contractions in English should be avoided in scientific article. Please remove "What's more" and revise the sentence accordingly.

Lines 302: Replace significant by “significantly”

Lines 314-315: Please add references.

Lines 368-369: Replace “One past study” by “Previous study”. Also, please provide references.

Lines 401-404: Please remove “Generally speaking” and revise the sentence accordingly.

Lines 412-430: The conclusion section is too large, please reduce the text and conclude concisely.

---

## [Author Response · Author response to Decision Letter 0]

13 Sep 2020

We really appreciate your earnest and careful review of our manuscript ID PONE-D-20-25310 entitled “First description and comparison of the morphological and ultramicro characteristics of the antennal sensilla of two fir longhorn beetles”. We also thank you very much for giving us an opportunity to consider again our revised paper.

Constructive comments from the reviewer was appreciated and carefully considered with a complete revision.

The response to manuscript revision instructions and reviewer’s comments point by point as listed below. Revised contents are showed by red fonts in the revised version.

Sincerely,

Zishu Dong

30 August, 2020

Response to Manuscript revision instruction:

Response: Thank you for your suggestion. In the revised version, revised contents is mainly shown in the following aspects:

(1)Revised contents are showed by red fonts in the revised version.

(2)We have checked carefully that our manuscript meets PLOS ONE's style requirements.

(3)We’ve integrated title page and main body together to make sure that the title page within our main document.

(4)Sorry for we forgot to provide a statement in our methods section that the works are feasibly. And we have added a statement in the methods section of manuscript and in the “Ethics Statement” field of the submission form. 

(5)For the acknowledgments section, we have revised according to your suggestion which avoided the appearance of any funding-related text in this manuscript. And we make sure that the funding information doesn’t appear in the Acknowledgments section or other areas of our manuscript. Besides, we have provided funding information that is currently declared in our Funding Statement. Moreover, we include the updated Funding Statement in your cover letter. Our final Funding Statement is “This research was funded by National Natural Science Foundation of China (31660626). The funders had no role in study design, data collection and analysis, decision to publish, or preparation of the manuscript”. We have deleted the study abroad fund program of Guangxi University in 2019. The cover letter also clearly states that the project only supported by National Natural Science Foundation of China (31660626). What's more, we also make a selection mark in the selection box of Current Funding Sources List of funding imformation.

(6)Corresponding author have completed the authorization process of ORCID iD.

Response to reviewer 1:

Comments 1: “As a whole，this article is tedious and repeated, such as the abstract, which should be compressed to more describe morphological and ultramicro characteristics”.

Response: Thanks for your suggestion. In the revised version, we rephrased our abstract (P. 2, L. 17-22) and conclusions (P. 26, L. 428-439) according to your comments.

Comments 2: “There are many logical problems in this paper, such as “the study on the comparison between…” (lines 316-319), “during field trials, C. villosulum shown…” (lines 368-371) and “this is related to the identification of…” (lines 378-383), move logically from one idea to the nextlogically from one idea to the next and don't skip steps, revise it again”.

Response: Thank you for your meaningful suggestions. After careful verification and cautious thinking, we revised the sentences respectively :

(1)At present, the study of antennae ultrastructure of longicorn beetles is focused on the comparison between male and female adults of each species. However, the interspecific comparison of antennae ultrastructure of different longicorn beetles with the same host plants is very rare (P. 22, L. 330-333). 

(2)The results of field trials show that C. villosulum shown specific attraction to the blend of 3-hydroxyhexan-2-one and the pyrrole, while A. asiaticus was only specifically attracted to the pyrrole as a single component (P. 24, L. 384-386). 

(3)This result suggests that these two sensilla very likely involved in the identification of sex pheromone components (P. 24, L. 395-396). 

Comments 3: “Overall, the authors paid attention to details in writing of the manuscript”.

Response: Thanks for your recognition of our work. What’s more, we are also particularly grateful to you for your careful review and appropriate evaluation of our article.

Response to reviewer 2:

Comments 1: “Lines 15-17: Please rewrite this sentence more clearly”.

Response: According to your suggestion. We rephrased the sentence into “ Four types (ten subtypes) of sensilla were both found on the antennae of these two fir longhorn beetles, named Böhm bristle (Bb), sensilla trichodea (ST I and II), sensilla basiconica (SB I, II and III), sensilla chaetica (SCh I, II, III and IV) ” (P. 2, L. 26-28).

Comments 2: “Lines 21-22: I think no need of this sentence”.

Response: We have deleted this sentence according to your comments (P. 2, L. 31).

Comments 3: “Lines 26-29: The given reference focus Chinese firs only in Fujian Province, China. Please provide references for their distribution in other countries or revised the statement which focus only China”.

Response: Thank you for your suggestion. We have replaced a new reference (P. 3, L. 39).

Earle CJ. Cunninghamia. The Gymnosperm Database. 2020 May 20. Available from: https://www.conifers.org/cu/Cunninghamia.php

Comments 4: “Lines 31: Replace indigenous by Indigenous”.

Response: We have revised it according to your suggestion (P. 3, L. 41).

Comments 5: “Lines 43-45: Please provide references for these sentences”.

Response: Sorry for my inappropriate expression. This is our own statement without reference. So, we change the sentence into “ To avoid its proliferation in other regions of the world, it is necessary to systematically study the two beetles in order to plan preventive measures such as developing detection tools” (P. 3, L. 43-45). 

Comments 6: “Lines 108: Please use full scientific name at the start of a sentence. Check whole MS and correct it”.

Response: Thank you for your reminder. We have checked whole MS, and just found the problem in this sentence. We changed this sentence into ‘There is a great difference between A. asiaticus and C. villosulum in the shape of the flagellum’ (P. 6, L. 119-120).

Comments 7: “Lines 110-114: The use of three adversative conjunction ‘Interestingly’, ‘Meanwhile’ and ‘Nevertheless’ led to a logic miss. Please consider to rewrite these three sentences. Also, please check whole MS regarding this kind of mistakes”.

Response: This is a meanful suggestion. We have rephrased this sentence as your comments (P. 6, L. 122-125). After checked whole MS, we also rephrased some sentence with similar problem (P. 22, L. 330-333; P. 24, L. 384-386; P. 24, L. 395-396).

Comments 8: “Lines 131-132: P should be italic. Correct this in whole MS”.

Response: We examined the whole MS carefully, and all p have been kept italic according to your comments ( P. 7, L. 142; P. 8, L. 143, 145-148; P. 9, L. 155, 158; P. 10, L. 160-162; P. 21, L. 314, 318).

Comments 9: “Lines 142-143: A. asiaticus and C. villosulum, should be italic”.

Response: We examined the whole MS carefully, and all latin names have been kept italic according to your comments (P. 6, L. 108-109; P. 9, L. 154-155).

Comments 10: “Lines 158: Remove ‘a large’ from sentence”.

Response: We have removed ‘a large’ from sentence according to your comments (P. 10, L. 171).

Comments 11: “Lines 159: Replace sex by ‘sexes’ ”.

Response: We have replaced ‘sex’ by ‘sexes’ according to your comments (P. 10, L. 172).

Comments 12: “Lines 177; 297-299; 378-379: The contractions in English should be avoided in scientific article. Please remove ‘What's more’ and revise the sentence accordingly”.

Response: Thank you for your suggestion. We have rephrased this sentence according to your comments (P. 11, L. 187-188). And we also have removed ‘What's more’ and revise the sentence accordingly (P. 21, L. 311-314; P. 25, L. 392-394).

Comments 13: “Lines 302: Replace significant by ‘significantly’ ”.

Response: We have replaced ‘significant’ by ‘significantly’ according to your comments (P. 21, L. 316).

Comments 14: “Lines 314-315: Please add references”.

Response: Thanks for your suggestion. We have added reference for this sentence (P. 22, L. 329).

Comments 15: “Lines 368-369: Replace ‘One past study’ by ‘Previous study’. Also, please provide references”.

Response: Thanks for your suggestion. We have rephrased this sentence according to your comments (P. 24, L. 382-384.)

Comments 16: “Lines 401-404: Please remove ‘Generally speaking’ and revise the sentence accordingly”.

Response: Thanks for your suggestion. We have rephrased this sentence according to your comments (P. 26, L. 417-419).

Comments 17: “Lines 412-430: The conclusion section is too large, please reduce the text and conclude concisely”.

Response: We rephrased our conclusion according to your comments (P. 26, L. 428-439).

---

## [Decision Letter · Decision Letter 1]

9 Oct 2020

First description and comparison of the morphological and ultramicro characteristics of the antennal sensilla of two fir longhorn beetles

PONE-D-20-25310R1

Dear Dr. Lu,

We’re pleased to inform you that your manuscript has been judged scientifically suitable for publication and will be formally accepted for publication once it meets all outstanding technical requirements.

Kind regards,

Yulin Gao

Academic Editor

PLOS ONE

---

## [Editor Report · Acceptance letter]

19 Oct 2020

PONE-D-20-25310R1 

First description and comparison of the morphological and ultramicro characteristics of the antennal sensilla of two fir longhorn beetles 

Dear Dr. Lu:

I'm pleased to inform you that your manuscript has been deemed suitable for publication in PLOS ONE. Congratulations! Your manuscript is now with our production department. 

Kind regards, 

on behalf of

Dr. Yulin Gao 

Academic Editor

PLOS ONE